# Latent Task-Specific Graph Network Simulators

**Philipp Dahlinger**[1] *  **Niklas Freymuth**[1]  **Michael Volpp**[2]
**Tai Hoang**[1]  **Gerhard Neumann**[1]
[1]Autonomous Learning Robots, Karlsruhe Institute of Technology, Karlsruhe
[2]Bosch Center for Artificial Intelligence, Renningen, Germany

## Abstract

Simulating dynamic physical interactions is a critical challenge across multiple scientific domains, with applications ranging from robotics to material science. For mesh-based simulations, Graph Network Simulators (GNSs) pose an efficient alternative to traditional physics-based simulators. Their inherent differentiability and speed make them particularly well-suited for inverse design problems. Yet, adapting to new tasks from limited available data is an important aspect for real-world applications that current methods struggle with. We frame mesh-based simulation as a meta-learning problem and use a recent Bayesian meta-learning method to improve GNSs adaptability to new scenarios by leveraging context data and handling uncertainties. Our approach, latent task-specific graph network simulator, uses non-amortized task posterior approximations to sample latent descriptions of unknown system properties. Additionally, we leverage movement primitives for efficient full trajectory prediction, effectively addressing the issue of accumulating errors encountered by previous auto-regressive methods. We validate the effectiveness of our approach through various experiments, performing on par with or better than established baseline methods. Movement primitives further allow us to accommodate various types of context data, as demonstrated through the utilization of point clouds during inference. By combining GNSs with meta-learning, we bring them closer to real-world applicability, particularly in scenarios with smaller datasets.

## 1 Introduction

Simulating physical systems is a fundamental challenge across a variety of scientific fields, with applications ranging from structural mechanics (Yazid et al., 2009; Zienkiewicz & Taylor, 2005; Stanova et al., 2015) over fluid dynamic (Chung, 1978; Zienkiewicz et al., 2013; Connor & Brebbia, 2013) to electromagnetism (Jin, 2015; Polycarpou, 2022; Reddy, 1994). Mesh-based simulations are often chosen for these tasks due to the computational efficiency and accuracy of the underlying finite element method (Brenner & Scott, 2008; Reddy, 2019). Yet, the diversity of modeled problems usually necessitates the development of task-specific simulators to accurately capture the relevant physical quantities (Reddy & Gartling, 2010). Despite these efforts, these specialized simulators can be slow and cumbersome, especially for larger simulations (Paszynski, 2016; Hughes et al., 2005).

As a result, data-driven models have gained traction as an appealing alternative (Guo et al., 2016; Da Wang et al., 2021; Li et al., 2022). Among these, general-purpose Graph Network Simulators (GNSs) have become increasingly popular (Battaglia et al., 2018; Pfaff et al., 2021; Allen et al., 2022b, 2023; Linkerhägner et al., 2023). Building on Graph Neural Networks (GNNs) (Scarselli et al., 2009; Wu et al., 2020; Bronstein et al., 2021), GNSs encode the simulated system as an interaction graph between nodes, predicting their dynamics. These models offer a significant speed advantage

---

*correspondence to `philipp.dahlinger@kit.edu`

NeurIPS 2023 AI for Science Workshop.

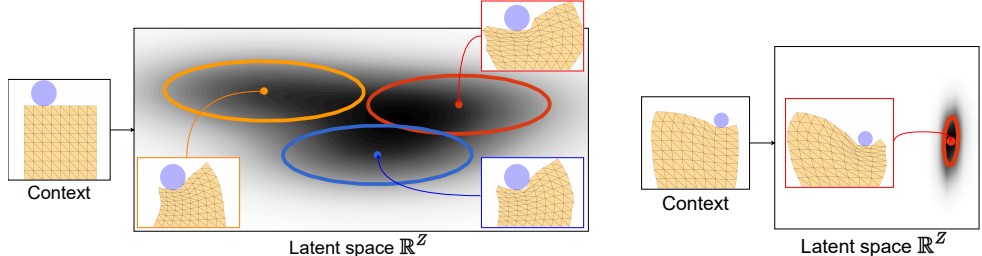

Figure 1: Illustration of the latent space of LTSGNS describing material properties of the mesh. (Left) Given an initial mesh as a context, the task posterior depicted as a black and white contour plot is spread out to include different possible mesh deformations. Thus, each sample of the posterior approximation (colored Gaussian components) results in a different yet plausible simulation outcome. (Right) When the material property can be inferred from additional context information, the posterior collapses to a unimodal distribution that represents deformations that are compatible with this context.

over classical simulators (Pfaff et al., 2021) while being fully differentiable, making them highly effective for applications like inverse design (Allen et al., 2022b; Xu et al., 2021).

GNSs are commonly trained through simple next-step supervision (Battaglia et al., 2018; Pfaff et al., 2021; Allen et al., 2023). During inference, entire trajectories are unrolled by iteratively predicting per-node dynamics from an initial system state. This process is susceptible to accumulating errors over time, especially as the input distribution diverges from the training set (Brandstetter et al., 2022; Han et al., 2022). While data augmentation strategies exist to offset this issue (Pfaff et al., 2021; Brandstetter et al., 2022), they neither correct mistakes once they have been made nor effectively address the challenges posed by partially known initial system states (Linkerhägner et al., 2023). Moreover, these models usually require large amounts of data to train, which is an issue in real-world scenarios where data is often sparse and the need for efficient adaptation to new tasks is crucial (Linkerhägner et al., 2023). In this work, we reformulate learned mesh-based simulation as a trajectory-level meta-learning problem that uses mesh states as a context set to address these limitations. We employ a Bayesian meta-learning approach based on non-amortized task posterior approximations (Volpp et al., 2021, 2023) for rapid adaptation to new task properties and uncertainties. We further mitigate the issue of error accumulation through the use of Probabilistic Dynamic Movement Primitivess (ProDMPs) (Schaal, 2006; Paraschos et al., 2013; Li et al., 2023) to represent higher-order dynamics of mesh nodes on a trajectory level. Combined, these methods allow us to model a distribution over unknown material properties and induced simulation trajectories from a given context set. When this context set contains sufficient information, the method is able to accurately determine specific system properties and adapt the simulation accordingly. We visualize an example in Figure 1. A more detailed overview of our approach, called Latent Task-Specific Graph Network Simulator (LTSGNS)[2] is shown in Figure 2.

We validate the effectiveness of LTSGNS on challenging deformable object simulations, showing superior prediction quality compared to MeshGraphNet (MGN), a state-of-the-art GNS. Furthermore, we showcase that our method can incorporate real-world observations such as point-clouds. This feature is particularly useful for applications dealing with sparse and incomplete data sets. The ability to accommodate such data types makes the model more applicable in practical scenarios and opens new avenues for research that require robust and generalizable graph network simulators.

## 2 Related Work

**Graph Network Simulators.** Recent research has increasingly focused on training deep neural networks for physical simulations as such models can yield significant speedups over traditional solvers while being fully differentiable (Pfaff et al., 2021; Allen et al., 2022a), making them a natural choice for e.g., model-based Reinforcement Learning (Mora et al., 2021) and Inverse Design problems (Baqué et al., 2018; Durasov et al., 2021; Allen et al., 2022a). Examples use Convolutional Neural Networks (CNNs) for fluid and aerodynamic flow simulations (Tompson et al., 2017; Guo

---

[2]Code available at `https://github.com/PhilippDahlinger/ltsgns_ai4science`

et al., 2016; Chu & Thuerey, 2017; Zhang et al., 2018; Bhatnagar et al., 2019; Ummenhofer et al., 2020; Abdolmaleki et al., 2016) and train either on a supervised loss or with the help of Generative Adversarial Networks (Goodfellow et al., 2014) in an adversarial fashion (Kim et al., 2019a; Xie et al., 2018). A popular class of learned neural simulators are GNSs (Battaglia et al., 2016; Sanchez-Gonzalez et al., 2020), a special type of GNN (Scarselli et al., 2009; Bronstein et al., 2021) that is designed to handle graph-structured physical data by modeling relations between arbitrary entities. Here, applications include particle-based simulations (Li et al., 2019; Sanchez-Gonzalez et al., 2020), atomic force prediction (Hu et al., 2021) and fluid dynamic problems (Brandstetter et al., 2022). Most notably, they have been applied to the mesh-based prediction of deformable objects (Pfaff et al., 2021; Weng et al., 2021; Han et al., 2022; Fortunato et al., 2022; Linkerhägner et al., 2023). Here, extensions include handling rigid objects (Allen et al., 2022b, 2023) and integrating learned adaptive mesh refinement strategies (Plewa et al., 2005; Yang et al., 2023; Freymuth et al., 2023) into the simulator (Wu et al., 2023). Closely related to our work, another extension utilizes additional sensory information to ground simulators to improve long-term predictions (Linkerhägner et al., 2023).

**Meta Learning.**   Meta-learning (Schmidhuber, 1992; Thrun & Pratt, 1998; Vilalta & Drissi, 2005; Hospedales et al., 2022) extracts inductive biases from a training set of related tasks in order to increase data efficiency on unseen tasks drawn from the same task distribution. In contrast to other multi-task learning methods, such as transfer learning (Krizhevsky et al., 2012; Golovin et al., 2017; Zhuang et al., 2020), which typically merely fine-tune or combine standard single-task models, meta-learning makes the multi-task setting explicit in the model architecture (Bengio et al., 1991; Ravi & Larochelle, 2017; Andrychowicz et al., 2016; Volpp et al., 2019; Santoro et al., 2016; Snell et al., 2017). This allows the resulting meta-models to learn *how* to learn new tasks with only a few context examples. A popular variant is the model-agnostic meta-learning (MAML) family (Finn et al., 2017; Grant et al., 2018; Finn et al., 2018; Kim et al., 2018), which employs standard single-task models and formulates a multi-task optimization procedure. The neural process (NP) model family (Garnelo et al., 2018a,b; Kim et al., 2019b; Gordon et al., 2019; Louizos et al., 2019; Volpp et al., 2021) is a complementary approach in the sense that it builds on a multi-task model architecture (Heskes, 2000; Bakker & Heskes, 2003), but employs standard gradient based optimization algorithms (Kingma & Ba, 2015b; Kingma & Welling, 2014; Rezende et al., 2014; Zaheer et al., 2017). Recently, Volpp et al. (2023) demonstrated the importance of accurate task posterior inference for efficient meta-learning by combining an NP-like architecture with more powerful inference and optimization schemes (Arenz et al., 2018; Lin et al., 2020; Arenz et al., 2023).

**Motion Primitives.**   Movement Primitives (MPs) (Schaal, 2006; Paraschos et al., 2013) allow for compact and smooth trajectory representation via a set of basis functions. Recent methods combine MPs with neural networks to increase their expressiveness (Seker et al., 2019; Bahl et al., 2020; Li et al., 2023). Among these, ProDMPs (Li et al., 2023) introduce a novel set of basis functions that sidestep an otherwise expensive numerical integration procedure in the training pipeline while being fully differentiable. Additionally ProDMPs can be queried at arbitrary points in time, making them particularly suitable for our approach.

## 3   Latent Task-Specific Graph Network Simulators

**Graph Network Simulators**   A MPN (Sanchez-Gonzalez et al., 2020; Pfaff et al., 2021) consists of a series of message passing steps which iteratively update latent node and edge features based on the graph topology. Given a graph $\mathcal{G} = (\mathcal{V}, \mathcal{E}, \mathbf{X}_\mathcal{V}, \mathbf{X}_\mathcal{E})$ with nodes $\mathcal{V}$, edges $\mathcal{E}$ and associated vector-valued node and edge features $\mathbf{X}_\mathcal{V}$ and $\mathbf{X}_\mathcal{E}$, each step is given as

$$\mathbf{h}_e^{k+1} = f_\mathcal{E}^k(\mathbf{h}_v^k, \mathbf{h}_u^k, \mathbf{h}_e^k), \quad \mathbf{h}_v^{k+1} = f_\mathcal{V}^k(\mathbf{h}_v^k, \bigoplus_{e=(v,u)} \mathbf{h}_e^{k+1}), \quad \text{with } e = (u, v) \in \mathcal{E}.$$

Here, $\mathbf{h}_v^0$ and $\mathbf{h}_e^0$ are embeddings of the system state per node and edge, and $\oplus$ is a permutation-invariant aggregation such as a sum, max, or mean operator. Each $f_\cdot^l$ is a learned function such as a small Multilayer Perceptron (MLP). The network's final output is a node-wise learned representation $\mathbf{h}$ that encodes local information about the graph topology and the predicted dynamics of the respective parts of the simulated system.

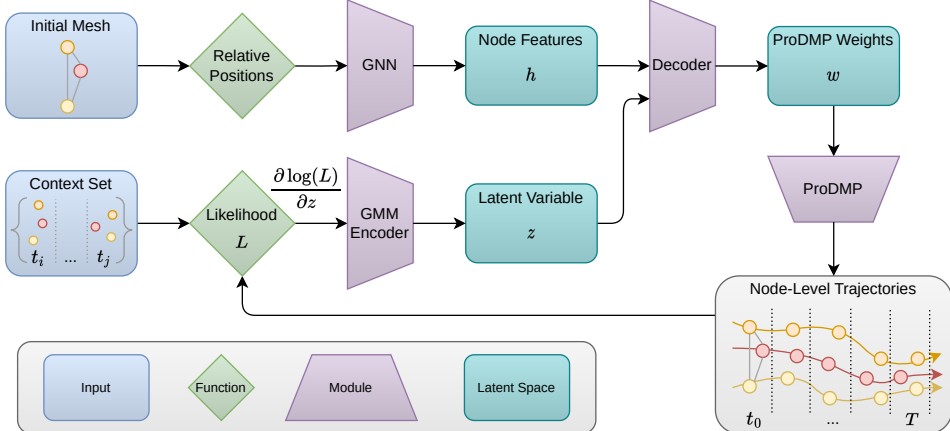

Figure 2: Schematic of the LTSGNS architecture. We compute relative node positions stored in the edges of initial mesh and obtain latent node features through a GNN. Combining the node features with a latent task-specific variable, we compute a trajectory per node using ProDMPs. To get the latent variable $z$ for the current context, we approximate the task posterior with a GMM. This requires the gradient of the likelihood with respect to $z$.

GNSs encode the system state as a graph, feed it through a Message Passing Network (MPN) and interpret the outputs per node as dynamics that can be used to forward the simulation in time using e.g., a forward-Euler integrator. The state encoding usually uses relative distances and velocities rather than absolute ones, as the resulting translation-invariance allows for better generalization over local areas (Sanchez-Gonzalez et al., 2020). When parts of the simulation are known, such as e.g., the positions of a robot's end-effector for a planned trajectory, only the remaining nodes are predicted. Existing GNSs usually minimize a next-step Mean Squared Error (MSE) per node during training and produce longer trajectories by iteratively applying the resulting forward dynamics (Pfaff et al., 2021). As this iterative dependence on previous predictions causes errors to accumulate over time, carefully tuned implicit de-noising strategies are often added during training (Sanchez-Gonzalez et al., 2020; Pfaff et al., 2021; Brandstetter et al., 2022). Here, we instead use ProDMPs to directly predict a compact trajectory representation per system node in a single step, reducing the effect of error accumulation similar to, e.g., temporal bundling (Brandstetter et al., 2022).

**Meta-Learning and Graph Network Simulators.** We view GNS as a meta-learning problem, where each task consists of simulating a deformable object with unknown material properties over time. Our goal is to learn a simulator which is adaptable to a specific scenario by observing context data. Following the notation of Volpp et al. (2023), the meta-dataset $\mathcal{D} = \mathcal{D}_{1:L}$ consists of simulation *tasks* $\mathcal{D}_l = \{\mathcal{G}_l, x_{l,1:T}, \boldsymbol{y}_{l,1:T}\}$ with an initial mesh $\mathcal{G}_l$ and time steps $x_{l,t}$ of the node positions $\boldsymbol{y}_{l,t} \in \mathbb{R}^{N \times D}$. We use $N$ as the number of nodes and $D$ for the world dimension of the simulation (usually $D = 3$ or $D = 2$). In contrast to standard meta-learning, we additionally use the initial graph of the system as a task-level context. Given the initial positions of the deformable mesh and rigid collider, the task is to predict the node positions $\boldsymbol{y}_{l,t}$ at time steps $x_{l,t}$. We note that the initial graph $\mathcal{G}_l$ does not contain the full system state, as we consider the material properties of the deformable object to be unknown.

**Model architecture.** Our model likelihood $L = p_{\boldsymbol{\theta}}(\boldsymbol{y}_{l,t} \mid x_{l,t}, \boldsymbol{z}_l, \mathcal{G}_l)$ is parametrized by a global parameter $\boldsymbol{\theta} \in \mathbb{R}^{d_{\boldsymbol{\theta}}}$ and defines the probability distribution over targets $\boldsymbol{y}_{l,t}$ at corresponding timesteps $x_{l,t}$, conditioned on a latent task descriptor $\boldsymbol{z}_l \in \mathbb{R}^Z$ and the initial graph $\mathcal{G}_l$, cf. Fig. 2. Given the graph $\mathcal{G}_l$, the initial mesh of a deformable object and a tractable rigid collider, we connect both objects based on proximity, compute relative distances between nodes and store this information in the edges. We then use a MPN to encode latent features $\boldsymbol{h} \in \mathbb{R}^H$ per node. We combine this encoding with the global latent variable $\boldsymbol{z}_l \in \mathbb{R}^Z \sim q_{\phi_l}(\boldsymbol{z}_l)$ by concatenating $\boldsymbol{z}_l$ to every node feature $\boldsymbol{h}$. Intuitively, the latent variable $\boldsymbol{z}_l$ encodes the material properties and high-level deformations of the respective

simulation, but does not focus on individual nodes. We use a simple MLP as the node-level decoder to yield final predictions per node.

Instead of iteratively predicting dynamics for the current simulation step like existing GNS (Pfaff et al., 2021; Allen et al., 2023), we use ProDMP (Li et al., 2023) to predict a representation of the full trajectory. The node-wise ProDMP weights $w \in \mathbb{R}^W$ define a trajectory over the full time horizon, and the ProDMP framework allows an efficient backpropagation from the trajectories to the weights $w$. As such, we can predict the node positions $\boldsymbol{y}_{l,t}$ at the desired time steps $x_{l,t}$ and use a node-wise Gaussian log-likelihood between the given and the predicted node positions to fit our latent space as detailed below.

**Model predictions.** Under our model, the predictive distribution for a target task $\mathcal{D}_*$, given a set of context examples $\mathcal{D}_*^c \subset \mathcal{D}_*$, is given by

$$p_{\boldsymbol{\theta}}(\boldsymbol{y}_{*,1:T} \mid x_{*,1:T}, \mathcal{D}_*^c) = \int \prod_{t=1}^{T} p_{\boldsymbol{\theta}}(\boldsymbol{y}_{*,t} \mid x_{*,t}, \mathcal{G}_*, \boldsymbol{z}_*) p_{\boldsymbol{\theta}}(\boldsymbol{z}_* \mid \mathcal{D}_*^c) \, \mathrm{d}\boldsymbol{z}_*, \tag{1}$$

where the *task posterior* distribution is given in terms of the model and a prior distribution $p(\boldsymbol{z}_*)$ over task descriptors by means of Bayes' theorem as

$$p_{\theta}(\boldsymbol{z}_* \mid \mathcal{D}_*^c) = \prod_{t=1}^{T^c} p_{\boldsymbol{\theta}}(\boldsymbol{y}_{*,t}^c \mid x_{*,t}^c, \boldsymbol{z}_*, \mathcal{G}_*) p(\boldsymbol{z}_*) \, / \, Z_*^c(\boldsymbol{\theta}) \equiv \tilde{p}_{\boldsymbol{\theta}}(\boldsymbol{z}_*) \, / \, Z_*^c(\boldsymbol{\theta}). \tag{2}$$

Here, $Z_*^c(\boldsymbol{\theta})$ denotes the marginal likelihood of the context data, i.e.,

$$Z_*^c(\boldsymbol{\theta}) \equiv p_{\boldsymbol{\theta}}(\boldsymbol{y}_{*,1:T^c}^c \mid x_{*,1:T^c}^c, \mathcal{G}_*) = \int \prod_{t=1}^{T^c} p_{\boldsymbol{\theta}}(\boldsymbol{y}_{*,t}^c \mid x_{*,t}^c, \boldsymbol{z}_*, \mathcal{G}_*) p(\boldsymbol{z}_*) \, \mathrm{d}\boldsymbol{z}_*. \tag{3}$$

For reasonably complex models, the marginal likelihood and, thus, the posterior distribution is intractable and requires approximation. As shown by Volpp et al. (2023), the predictive accuracy is highly dependent on the accuracy of the *task posterior approximation*, causing us to mimic their approach and employ an expressive full-covariance Gaussian Mixture Model (GMM) of the form

$$p_{\boldsymbol{\theta}}(\boldsymbol{z}_* \mid \mathcal{D}_*) \approx q_{\boldsymbol{\phi}_*}(\boldsymbol{z}_*) = \sum_{k=1}^{K} w_{*,k} \mathcal{N}(\boldsymbol{z}_* \mid \boldsymbol{\mu}_{*,k}, \boldsymbol{\Sigma}_{*,k}). \tag{4}$$

We use $K$ mixture components with corresponding weights $w_{*,k}$, means $\boldsymbol{\mu}_{*,k}$, and covariance matrices $\boldsymbol{\Sigma}_{*,k}$, which we collective denote by $\boldsymbol{\phi}_*$. To fit the variational distribution $q_{\boldsymbol{\phi}_*}(\boldsymbol{z}_*)$, we also follow Volpp et al. (2023) and use the Trust Region Natural Gradient Variational Inference (TRNG-VI) method, specifically SEMTRUX (Arenz et al., 2023). This requires samples of $\nabla_z \tilde{p}_{\boldsymbol{\theta}}(\boldsymbol{z}_*)$, which can be readily obtained using standard automatic differentiation tools (Paszke et al., 2019).

**Meta-training.** The aim of meta-learning is to automatically encode inductive biases towards the task distribution extracted from the meta-dataset $\mathcal{D}$ in the task-global parameter $\boldsymbol{\theta}$. To this end, we maximize w.r.t. $\boldsymbol{\theta}$ the log marginal likelihood, which is given as the sum of the per-task log marginal likelihoods

$$\log Z_l(\boldsymbol{\theta}) \equiv \log p_{\boldsymbol{\theta}}(\boldsymbol{y}_{l,1:T} \mid x_{l,1:T}, \mathcal{G}_l) = \log \int \prod_{t=1}^{T} p_{\boldsymbol{\theta}}(\boldsymbol{y}_{l,t} \mid x_{l,t}, \boldsymbol{z}_l, \mathcal{G}_l) p(\boldsymbol{z}_l) \, \mathrm{d}\boldsymbol{z}_l. \tag{5}$$

As discussed above, the marginal likelihood is intractable, which is why we employ an evidence lower bound (ELBO) of the form

$$\mathrm{ELBO}_l(\boldsymbol{\theta}) = \mathbb{E}_{q_{\boldsymbol{\phi}_l}(\boldsymbol{z}_l)} \left( \sum_{t=1}^{T} \log p_{\boldsymbol{\theta}}(\boldsymbol{y}_{l,t} \mid x_{l,t}, \boldsymbol{z}_l, \mathcal{G}_l) + \log \frac{p(\boldsymbol{z}_l)}{q_{\boldsymbol{\phi}_l}(\boldsymbol{z}_l)} \right) \leq \log Z_\ell(\boldsymbol{\theta})$$

as a surrogate objective for maximization of the log marginal likelihood. The efficiency of this optimization scheme increases with the tightness of the lower bound, which is in turn controlled by the task posterior approximation quality of the variational distribution $q_{\boldsymbol{\phi}_l}$ (Bishop, 2006; Volpp et al., 2023). Therefore, we also employ the expressive GMM-TRNG-VI approximation procedure to fit $q_{\boldsymbol{\phi}_l}$ during meta-training. The resulting ELBO can then be efficiently approximated using a Monte-Carlo estimation of the expectation, and be optimized using standard gradient-based optimization (Kingma & Welling, 2013; Rezende et al., 2014; Volpp et al., 2023; Kingma & Ba, 2015b). After meta-training, we fix the parameters $\boldsymbol{\theta}$ and use them for predictions on unseen tasks.

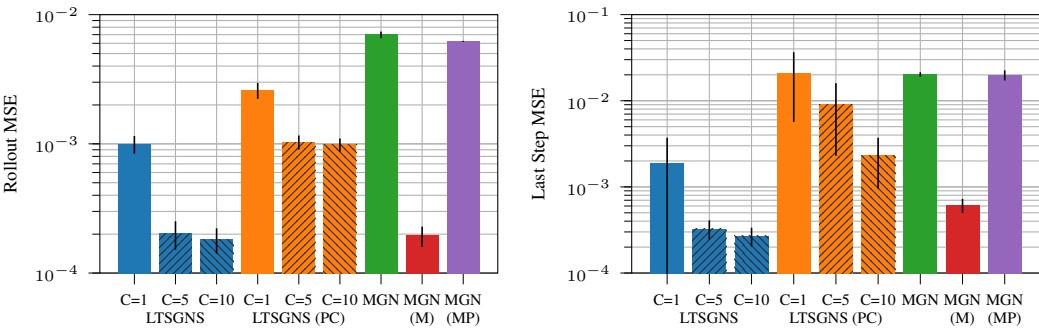

Figure 3: (Left) Rollout and (Right) Last Step MSE of the different methods on the *Deformable Plate* task. Both metrics are plotted on a logarithmic scale. LTSGNS outperforms MGN and its MP variant for both metrics from a single context point, further improving its performance when provided with additional context information. During inference, point cloud information can be used as a context set, making for easier context acquisition at the cost of slightly worse predictions. Overall, the performance metrics for both Rollout and Last Step MSE are fairly similar. However, the Last Step MSE exhibits increased variance across all methods, making it less consistent in comparison.

## 4 Experiments

**Setup.** We base our experimental setup on that of Linkerhägner et al. (2023), adapting it to meta-learning and the prediction of full trajectories using movement primitives. To represent the system as a graph, we employ one-hot encoding to differentiate between deformable objects and colliders, and encode relative distances between neighboring nodes in their edges. We do not use explicit world edges, but include the distances between nodes in mesh space. We utilize the context set $\mathcal{D}_*^c$, comprising the initial mesh and $C$ randomly sampled simulation states, to predict node-wise ProDMP parameters $w$. These parameters encode the full mesh trajectory, and can be queried to obtain the system state at arbitrary timesteps.

All models are optimized using Adam (Kingma & Ba, 2015a) with a learning rate of $5 \times 10^{-4}$. For the MPNs, we use 5 separate message passing steps for all methods. For LTSGNS, we repeat each step 2 times in an inner loop to increase the receptive field of the individual nodes, as there is no iterative prediction that could otherwise pass implicit information. Each block uses a latent dimension of 128, LeakyReLU activations and a 1-layer MLPs for its node and edge updates. We additionally apply Layer Normalization (Ba et al., 2016) and Residual Connections (He et al., 2016) independently for both node and edge updates. We repeat each experiment for 5 random seeds and report the mean and standard deviation over these seeds. Each run's results are averaged over all trajectory steps and test set trajectories. We evaluate the models using the *Rollout MSE*, which is the average MSEs of all simulation steps, and the *Last Step MSE*, which is the error of the final simulation step. Additional details on our experimental setup are provided in Appendix A.

**Tasks.** We consider a simpler 2-dimensional *Deformable Plate* and a more challenging 3-dimensional *Tissue Manipulation* task (Linkerhägner et al., 2023). Both tasks use Simulation Open Framework Architecture (SOFA) (Faure et al., 2012) to generate the underlying ground truth data, and use triangular surface meshes for the simulation. While the initial meshes are known, both tasks use materials with a randomized and unknown Poisson's ratio (Lim, 2015) that governs whether the material contracts or expands under deformation. The *Deformable Plate* task simulates different trapezoids that are deformed by a circular collider with constant velocity and varying size and starting position. Each trajectory consists of a mesh with 81 nodes that is deformed over 50 timesteps, and we use 675/135/135 trajectories for training, evaluation and testing respectively. The *Tissue Manipulation* task simulates a common scenario in surgical robotics where a piece of tissue is deformed by a gripper. Here, the gripper starts attached to a random position of the object and moves in a random direction with constant velocity. The simulated mesh has 361 nodes, and we use 600/120/120 training, evaluation and testing trajectories with 100 steps each. All tasks are normalized to be in $[-1, 1]$.

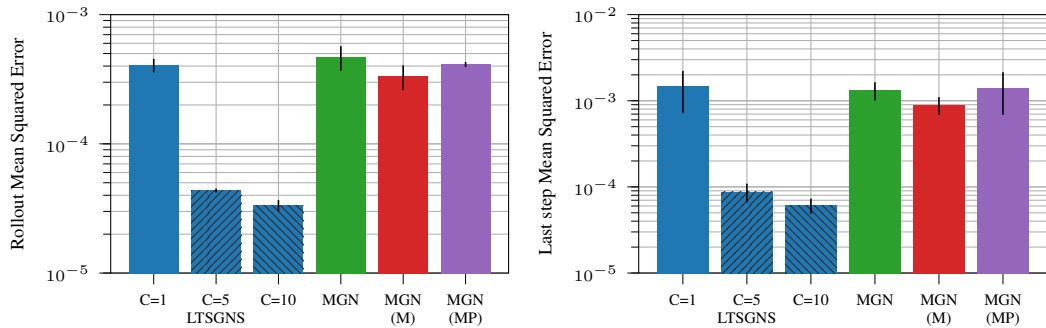

Figure 4: (Left) Rollout and (Right) Last Step MSE of the different methods on the *Tissue Manipulation* task. Both metrics are plotted on a logarithmic scale. LTSGNS yields more accurate simulations with increasing context size, clearly outperforming the MGN baselines for 5 or more context points.

**Baselines and Ablations.** We use MGN (Pfaff et al., 2021) as our main baseline. MGN iteratively predicts the velocities of the current simulation step to generate the next mesh state. It is trained to minimize the 1-step MSE over node velocities and crucially employs Gaussian input noise (Brandstetter et al., 2022) to prevent error accumulation over time and thus generalize from 1-step predictions to larger rollouts during inference. We follow previous work (Linkerhägner et al., 2023) and set the standard deviation of the input noise to $0.01$. We experiment with both MGN without information about the material properties, and with a variant that includes this additional information, called MGN(M). The latter renders the simulation deterministic with respect to the initial system state and sets an upper performance limit for the standard MGN model. Additionally, we compare to an MGN(MP) variant that incorporate ProDMPs to directly predict a trajectory for each node feature instead of iteratively predicting the next state. Since this method directly predicts the full trajectory, it does not use Gaussian input noise. Building on our model's adaptability to different context sets, we conduct an ablation experiment focused on practical applications. Specifically, we test the use of point clouds as the context set during inference. This is particularly relevant as point clouds can be readily generated from depth cameras in real-world settings, while mesh states cannot. Importantly, this adjustment requires no modifications to the existing training process.

**Results.** Figure 3 shows results for the *Deformable Plate* task. We find that LTSGNS outperforms MGN even when provided with just a single context point. The model's performance continues to improve as the size of the context set increases. Specifically, for a context set with 10 points, LTSGNS outperforms MGN(M) even though the latter has direct access to the ground truth material information. Although the performance drops when using point cloud data as context instead of system states, LTSGNS still outperforms MGN and benefits from the addition of more contextual information. Similarly, Figure 4 evaluates the *Tissue Manipulation* task. Here, LTSGNS again improves with an increasing context size, outperforming all MGN baselines for 5 or more context points. Providing material properties still improves MGN, but the difference is less significant than for the *Deformable Plate* task, presumably because the dynamics are overall harder to predict even with this additional information. We provide additional results for larger context sets in Appendix B.1.

Figure 5 visualizes exemplary final simulation steps, supporting the findings above. LTSGNS accurately simulates the object's deformation from a single context point, and further improves when provided with additional context information. Opposed to this, MGN, even when provided with explicit material properties or when combined with ProDMPs, fails to produce consistent meshes. Appendix B.2 shows visualizations for full rollouts.

## 5 Conclusion

We introduce LTSGNS, a novel Graph Network Simulator that employs meta-learning and movement primitives for accurate probabilistic predictions in physical simulations. Our model uses meta-learning and movement primitives to effectively addresses the issue of error accumulation and dynamically adapt to context information during inference. LTSGNS is also able to accommodate sensory inputs like point clouds during inference, broadening its applicability in real-world scenarios. We

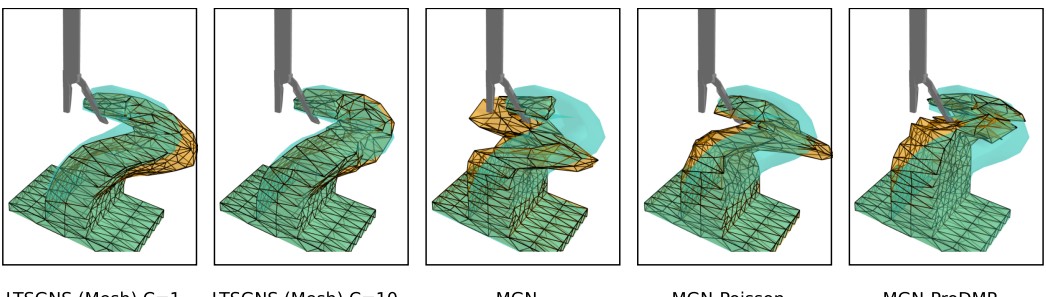

| LTSGNS (Mesh) C=1 | LTSGNS (Mesh) C=10 | MGN | MGN-Poisson | MGN-ProDMP |

Figure 5: Final simulation step of an exemplary test trajectory for the *Tissue Manipulation* task across different methods. Blue denotes the ground position of the deformable object, while the wireframe and yellow shading outline the predicted mesh. Only LTSGNS provides accurate predictions, which further improve as the size of the context set increases.

experimentally validate the effectiveness of our approach compared to existing baselines, particularly in tasks involving uncertain material properties. The model's ability to produce distributions over trajectories adds an extra layer of robustness, making it a valuable tool for both academic research and practical applications where data is sparse.

**Limitations and Future Work**    We currently consider each trajectory as a task, and require data from either states or point clouds of this trajectory to fit our model during inference. However, as generating such data is often impractical in real-world scenarios, we plan to instead group different material properties into tasks. This shift will enable the model to encode abstract material behavior, without being tied to a particular mesh topology or simulation. We further aim to extend our approach to accommodate longer simulations to fully capitalize on the benefits of incorporating movement primitives. Combining both, we plan to apply our model to real-world deformations. Here, an additional focus will be on integrating online re-planning of trajectories, thereby enhancing prediction accuracy when live sensory information is available.

# 6    Acknowledgments

This work was funded by the Deutsche Forschungsgemeinschaft (DFG, German Research Foundation) – Project 5339. This work was supported by funding from the pilot program Core Informatics of the Helmholtz Association (HGF). The authors acknowledge support by the state of Baden-Württemberg through bwHPC, as well as the HoreKa supercomputer funded by the Ministry of Science, Research and the Arts Baden-Württemberg and by the German Federal Ministry of Education and Research.

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

# A  Experimental Protocol

In order to promote reproducibility, we offer comprehensive information regarding our experimental methodology. In Table 1, we give our used hyperparameters for the experiments. To obtain a more detailed description of the tasks and the dataset, please refer to Linkerhägner et al. (2023) Appendix B.

In this paper, we use a training approach that adapts its batch size dynamically based on the context set size. To achieve this, we define a batch cost as an upper bound, calculated as 0.8 plus 0.2 times the batch size. The batch size is then meticulously chosen to align with the specified batch cost, allowing us to optimize our training process effectively. Specifically, for the deformable plate task, we set a target total batch cost of 60, whereas for the tissue manipulation task, we aim for a batch cost of 300. The training for the deformable plate task was conducted on an NVIDIA GeForce RTX 3080 GPU, while the tissue manipulation task utilized an NVIDIA A100 GPU for efficient processing and optimization. The availability of more powerful hardware enabled us to opt for a larger batch cost in the context of the tissue manipulation task.

For each trajectory, we generate 30 auxiliary training tasks, as inspired by the approach outlined in Volpp et al. (2023), Appendix A3.2. In this process, we randomly select pairs of $(x_{l,t}, \boldsymbol{y}_{l,t})$ for each auxiliary training task. Additionally, we designate one of the first 30 time steps as the time steps for constructing the graphs denoted as $\mathcal{G}_l$. During the testing phase, the tasks consistently make use of the initial mesh for the corresponding $\mathcal{G}_l$, and we adapt the context size $C$ differently for each evaluation scenario.

Table 1: Table listing the hyperparameters and configurations of the experiments

| Parameter | Value |
|---|---|
| Batches per epoch | 500 |
| Epochs | 1000 |
| Node feature dimension | 128 |
| Latent variable dimension | 8 |
| Decoder dimension (deformable plate) | 128 |
| Decoder dimension (tissue manipulation) | 256 |
| Message passing blocks | 5 |
| Message passing repeats (MGN) | 1 |
| Message passing repeats (LTSGNS) | 2 |
| MPN Aggregation function | Mean |
| Learning rate | $5.0e-4$ |
| Gaussian likelihood standard deviation | 0.01 |
| Number of $z$ samples for ELBO estimation | 32 |
| Number of GMM components | 3 |
| Auxiliary train tasks per trajectory | 30 |
| Activation function | Leaky ReLU |
| SEMTRUX component KL bound | 0.01 |
| Number of ProDMP basis functions | 10 |

# B  Additional Results

## B.1  Evaluations.

We additionally show how LTSGNS performs for larger context sizes in Figures 6 (*Deformable Plate*) and 7 (*Tissue Manipulation*). While the inclusion of more context information generally enhances performance, it sometimes diminishes performance for the Last Step MSE when point clouds are used as the context. We hypothesize that this effect occurs because a larger context size shifts the balance between the likelihood $L$ and the prior $p(z_l)$, thereby magnifying any existing inaccuracies in the model that may exist in more complex tasks.

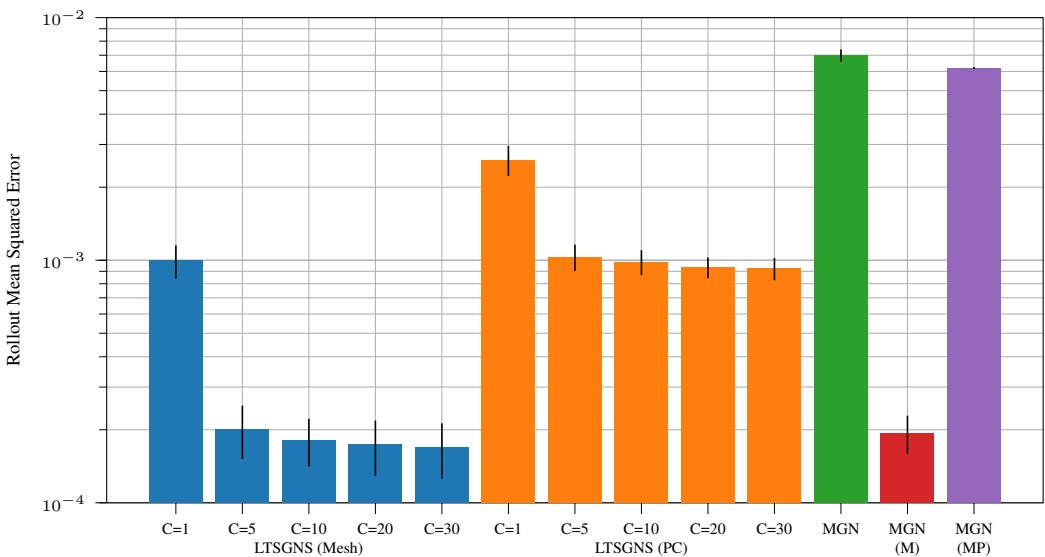

Figure 6: (Left) Rollout and (Right) Last Step MSE for larger context sizes ($C = 20$, $C = 30$) compared to the methods presented in Figure 3 for the *Tissue Manipulation* task.

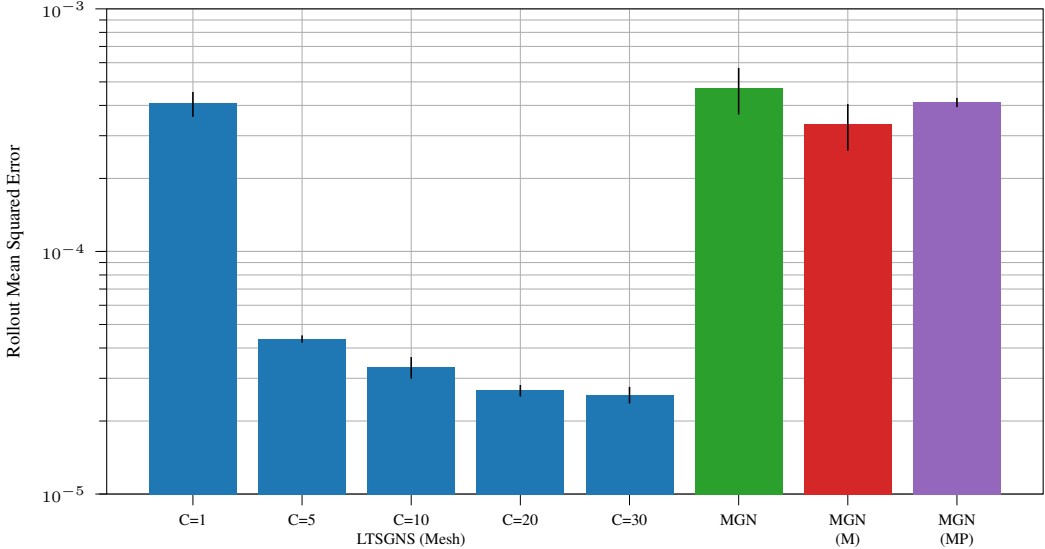

Figure 7: (Left) Rollout and (Right) Last Step MSE for larger context sizes ($C = 20$, $C = 30$) compared to the methods presented in Figure 4 for the *Tissue Manipulation* task.

## B.2 Visualizations.

We provide additional visualizations for all methods over different timesteps. Figure 6 shows a test trajectory for the *Deformable Plate* task, and Figure 9 visualizes the same for the *Tissue Manipulation* task. Both figures show that LTSGNS performs similar to MGN for a single context point, but significantly improves performance when provided with additional information. Especially for $C = 10$ data points in the context set, the predictions are visually consistent with the ground truth.

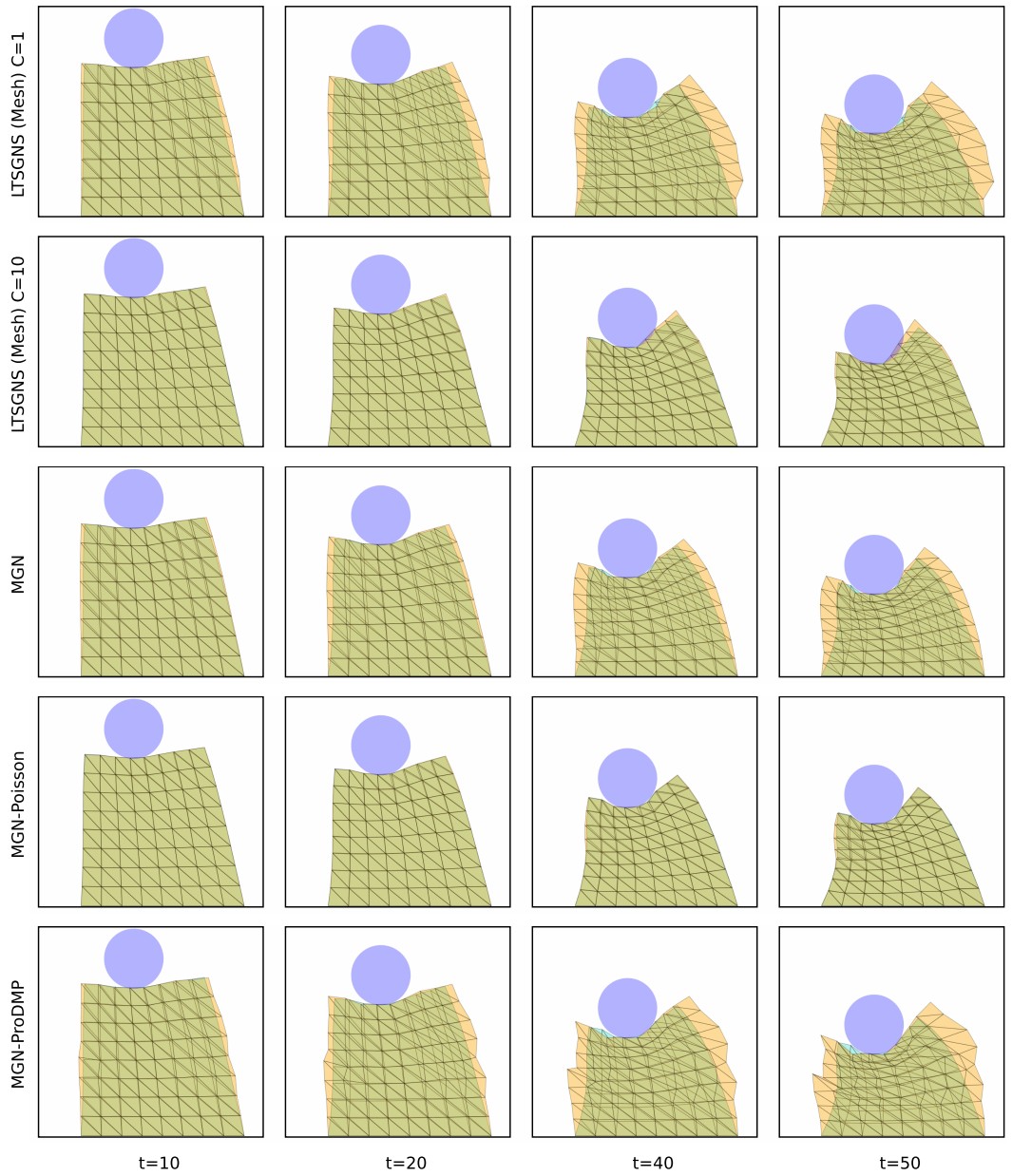

Figure 8: Simulation over time of an exemplary test trajectory of the *Deformable Plate* task across different methods. Blue denotes the ground position of the deformable object, while the wireframe and yellow shading outline the predicted mesh.

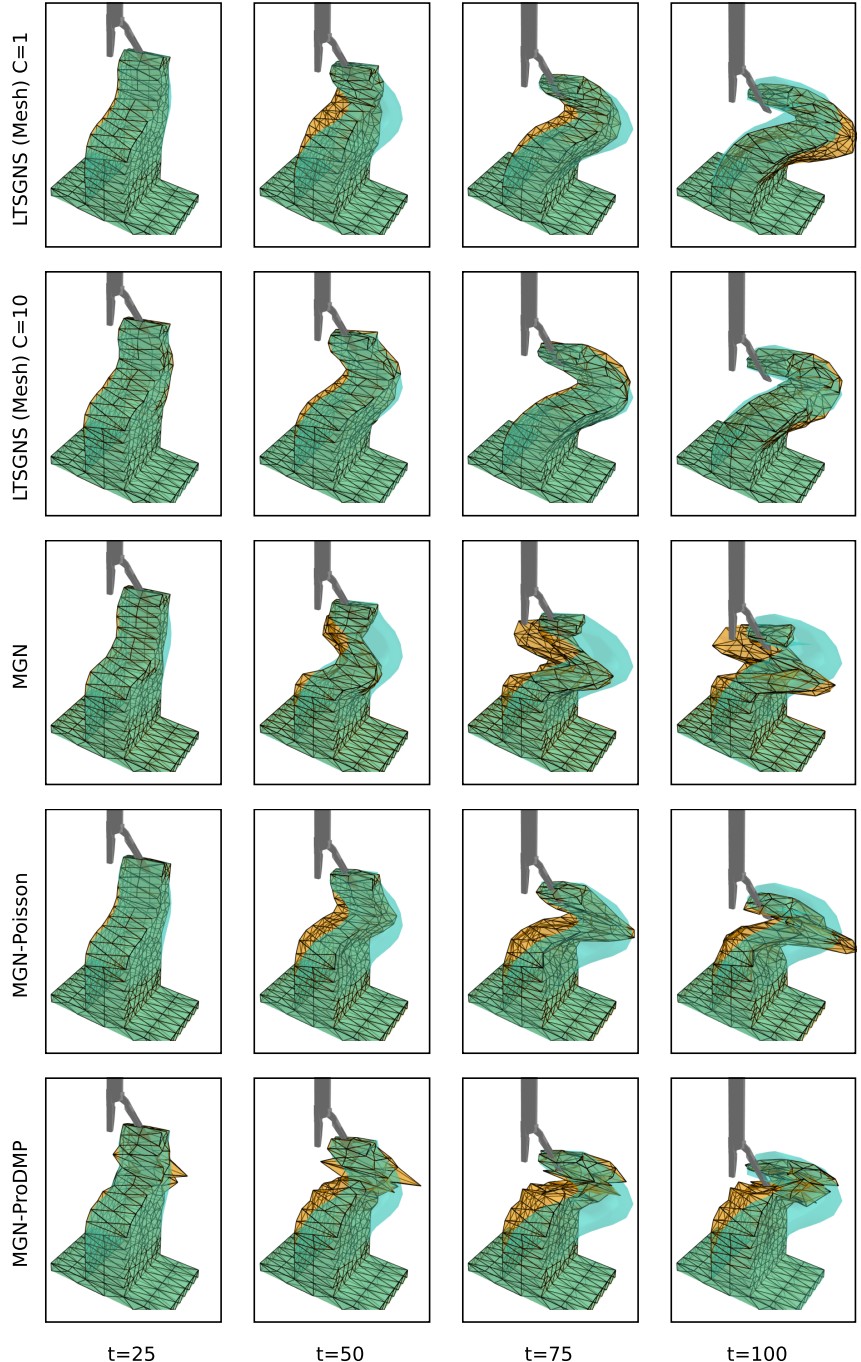

Figure 9: Simulation over time of an exemplary test trajectory of the *Tissue Manipulation* task across different methods. Blue denotes the ground position of the deformable object, while the wireframe and yellow shading outline the predicted mesh.

