# OpenReview forum: "Latent Task-Specific Graph Network Simulators"
_NeurIPS.cc/2023/Workshop/AI4Science — NeurIPS2023-AI4Science Poster_

### Official Review · Reviewer_1dpj · 2023-10-17
**Interesting meta-learning approach to Graph Network Simulation**

**Rating:** 7
**Confidence:** 4

**Review:**

**Strengths**

**S1:** The authors did a great job in concisely explaining the modeling approach of LTSGNS step-by-step while assuming only minimal prior knowledge of meta learning and variational inference. This is particularly helpful since the audience of this work is the broader learning-to-simulate community, where not everybody may have extensive background in these fields. More generally, the paper is easy to follow and enjoyable to read for the target audience.

**S2:** The results demonstrate that the proposed method has the potential to outperform MGN, a popular mesh-based GNS, even when MGN is conditioned on the material properties directly. This is a promising finding and warrants further research and experiments.

**S3:** The accumulation of error inherent to autoregressive neural simulators is a problem that is notoriously difficult to solve, and no real satisfying solutions exist. This paper proposes to use Probabilistic Dynamic Movement Primitives (ProDMPs) to address this issue. Rather than iteratively ‘integrating’ over time to simulate a trajectory, this approach predicts a representation of the entire trajectory in a single step, sidestepping error accumulation issues, which is relevant to the broader community. The decoupling of the numerical integration from the learning pipeline could also make training more scalable compared to training strategies that require iteratively applying the model at training time. However, see also comment C3 on this aspect.

**Comments**

**C1:** It would be great if a mathematical description of ProDMPs is included in the paper to make the reading more self-contained. Ideally, this should be in the main text, even if some other content needs to be sacrificed.

**C2:** The context sets that have been explored in the experiments are sets of randomly sampled simulation states from the trajectory. However, in many practical settings only the initial condition will be available. Although the authors mention grouping different material properties into tasks in the future work section, as opposed to defining tasks at the single-trajectory level, this somewhat limits the applicability of the approach as currently presented in the paper. Moreover, except for the part of the trajectory beyond the last state that has been sampled, this effectively changes the problem formulation into an ‘interpolation’ task (as in interpolation between the time points of the sampled states), rather than the task of ‘extrapolating’ an entire trajectory from the initial condition. In this light, it is not clear how figures 3 and 4 should be interpreted when comparing LTSGNS with MGN.

**C3:** The authors claim that using ProDMPs in their model mitigates the accumulation of errors. While I see that conceptually this seems plausible, I find the experimental support for this claim lacking. Sure, the last-step and rollout MSEs are reported, but given that the use of ProDMPs to mitigate rollout errors is presented as one of the main contributions of the work, I would have expected a more rigorous evaluation of rollout stability, e.g. plots of the MSE over time, what percentage of runs remain stable over time, or similar. In addition, it would be good to include results of an ablation of LTSGNS without the ProDMP component to assess the benefits of ProDMP, especially since the presented results seem to indicate that incorporating ProDMP in the MGN model does not seem beneficial for performance. Adding this ablation would help members of the community in judging whether they should consider integrating ProDMP-like methods into their own approaches, potentially enhancing the impact of this paper.

**Minor comments and typos**

- line 26: 'the' should be removed
- line 154: should this be ‘[…] but does _not_ focus on individual nodes.’?
- Eq. 3: I think the $\log$ should not be there

**Conclusion**

I enjoyed reading this paper, which is certainly relevant for the AI4Science community. The main technical contributions are twofold: one the one hand, the Bayesian meta-learning approach is inspiring as it could lead to the development of GNS models that generalize over tasks; on the other hand, the integration of ProDMPs into the model architecture may be beneficial in improving performance of GNS models in general, especially over long rollout horizons, which could benefit the whole community. Combined, these contributions make that the presented method is technically solid and interesting.
I do have some concerns on the experimental evaluation and to what extent some of the claims are supported by the results -- see comments C2 and C3. I encourage the authors to address these concerns. However, I believe that for this workshop the merits of the innovative approach substantially outweigh the limitations of the experimental evaluation. Therefore, I recommend to accept the paper.

---

### Official Review · Reviewer_26UQ · 2023-10-25
**Authors have proposed a meta-learning based graph neural network model to simulate the dynamics of physical proess of material system.**

**Rating:** 7
**Confidence:** 3

**Review:**

Authors have proposed a meta-learning based graph neural network (GNN) model to simulate the dynamics of material system such as deformation behavior of a plate over time. Basically the GNN learn to predict trajectory of material system over time, where the proposed meta-learning model helps the GNN model to learn new tasks quickly by utilizing the knowledge acquired in the  meta-learning process over wide range of prior tasks.

---

### Meta-Review · Area_Chair_18fv · 2023-10-26

**Recommendation:** Accept (Poster)
**Confidence:** 4

**Metareview:**

In this paper, authors have presented an interesting formulation of mesh-based simulation as a meta-learning problem and applied a recent Bayesian meta-learning algorithm to improve Graph Network Simulators. Additionally, this paper proposes to use Probabilistic Dynamic Movement Primitives to address accumulation of error inherent to autoregressive neural simulators.
Both reviewers agree on contributions and importance of the paper. One of the reviewer raises concerns on experimental evaluations, however, the technical contributions of paper outweight limitations. So, I recommend acceptance.